# Overexpression of a Novel Noxo1 Mutant Increases Ros Production and Noxo1 Relocalisation

**DOI:** 10.3390/ijms24054663

**Published:** 2023-02-28

**Authors:** Fatima-Zahra Benssouina, Fabrice Parat, Claude Villard, Ludovic Leloup, Françoise Garrouste, Jean-marc Sabatier, Lotfi Ferhat, Hervé Kovacic

**Affiliations:** Aix-Marseille Univ, CNRS, INP, Inst Neurophysiopathol, 13005 Marseille, France

**Keywords:** Nox1, NADPH oxidase, Noxo1, Rac1, proteasome, D-Box, reactive oxygen species

## Abstract

Noxo1, the organizing element of the Nox1-dependent NADPH oxidase complex responsible for producing reactive oxygen species, has been described to be degraded by the proteasome. We mutated a D-box in Noxo1 to express a protein with limited degradation and capable of maintaining Nox1 activation. Wild-type (wt) and mutated Noxo1 (mut1) proteins were expressed in different cell lines to characterize their phenotype, functionality, and regulation. Mut1 increases ROS production through Nox1 activity affects mitochondrial organization and increases cytotoxicity in colorectal cancer cell lines. Unexpectedly the increased activity of Noxo1 is not related to a blockade of its proteasomal degradation since we were unable in our conditions to see any proteasomal degradation either for wt or mut1 Noxo1. Instead, D-box mutation mut1 leads to an increased translocation from the membrane soluble fraction to a cytoskeletal insoluble fraction compared to wt Noxo1. This mut1 localization is associated in cells with a filamentous phenotype of Noxo1, which is not observed with wt Noxo1. We found that mut1 Noxo1 associates with intermediate filaments such as keratin 18 and vimentin. In addition, Noxo1 D-Box mutation increases Nox1-dependent NADPH oxidase activity. Altogether, Nox1 D-box does not seem to be involved in Noxo1 degradation but rather related to the maintenance of the Noxo1 membrane/cytoskeleton balance.

## 1. Introduction

Reactive oxygen species (ROS) serve as a second messenger in cell signaling, and their regulation is finely controlled. However, uncontrolled generation of ROS by mitochondria or by NAPDH Oxidase complex hyperactivation leads to damage to intracellular molecules, such as lipids, DNA, and proteins [1]. This deregulation is classically observed during inflammation processes or diseases such as cancer or neurodegenerative diseases [2]. Therefore, the generation of ROS must be tightly regulated [3].

NADPH oxidases represent a family of enzymes whose main and primary function is to produce ROS. This family is composed of seven different NADPH oxidase enzyme complexes (Nox1 to 5; Duox1 and Duox2) [4]. Phosphorylation of the Nox2 complex has long been identified as a regulator of the oxidative burst in neutrophils [5]. Nox2, as an active ROS-producing center, is activated by its association with the other subunits of the NADPH oxidase complex (p47phox, p67phox, and Rac). Phosphorylation of p47phox by protein kinases C (PKC) results in exposure of the tandem SH3 domain in the central region of p47phox, a domain that is masked when the protein is unphosphorylated by a folding of the AIR (auto-inhibitory region) sequence of p47phox. Once unmasked, the SH3 sequence of p47phox can interact with the proline-rich region (PRR) of p22phox and a PRR in the COOH-terminal region of p47phox for binding to the SH3 domain of p67phox, leading to activation of the Nox2 complex [6].

Regarding the activation of Nox1, the activator Noxa1 homologous to p67phox and the organizer Noxo1 homologous to p47phox have been identified as regulatory subunits [7]. For a long time, it was assumed that Nox1 activity was constitutive due to the absence of an AIR sequence in the Noxo1 protein resulting in its constant binding to Nox1. However, many studies have shown transient activations of Nox1, suggesting a fine regulation of the activity of the complex. Several regulatory mechanisms have been proposed and identified. First, phosphorylation of Noxa1 via protein kinase A [8] represents a mode of deactivation of Nox1 by blocking the interaction of Noxa1 with Nox1 and Rac1 [9], while other phosphorylation of Noxa1 seems to be activating [10]. Phosphorylation of serine (Ser) 154 of Noxo1 has been identified as activating Nox1 activity [11], while phosphorylation of threonine (Thr) 341 is required for phorbol myristate acetate activation [12]. More recently, the protein kinase casein kinase 2 has been shown to phosphorylate different sites at the C-terminus of the Noxo1 gamma isoform (phosphorylation sites corresponding to Ser-368, Thr-373, and Thr-374). These phosphorylations lead to an inhibition of Nox1 activation [7]. Thus, there are many regulatory phosphorylations of Noxo1 and Noxa1. Besides phosphorylations, modification of the stability and half-life of proteins represents another way to regulate Nox1 activity. Protein ubiquitination forms a molecular marker for proteasome degradation [13]. Ubiquitination is catalyzed by a cascade of three enzymes involving the *ubiquitin* (Ub) activating enzyme E1, the Ub conjugating enzyme E2, and the Ub E3 ligase. The E3 ligases represent a family of more than 600 different proteins that all have specific substrates [14]. The specificity of E3 ligases for their substrate has led to the search for motifs that promote these interactions, the degrons. Known degrons are either short amino acid sequences, structural motifs, or exposed amino acids located anywhere in the target protein [15]. Proteins can also contain several degrons. In previous work, we had shown that the GTPase Rac1 was addressed to degradation by the proteasome when it was in GTP active form upon activation of NADPH oxidase. We, therefore, proposed proteasomal regulation as an additional means of controlling NADPH oxidase Nox1 activity [16]. The E3 ubiquitin ligase HACE1 (HECT Domain and Ankyrin Repeat Containing E3 Ubiquitin Protein Ligase 1) was identified as the ligase responsible for the ubiquitination of active Rac1 [17]. More recent works have shown that Noxo1 is addressed and degraded at the proteasome [18,19]. These studies identify that deubiquitinase cylindromatosis (CYLD) increases Noxo1 ubiquitination and the Casitas B-lineage lymphoma (Cbl) E3 ligase, a RING finger E3 ligase, as the ubiquitinating enzyme of Noxo1. We thus try to identify some potential degron in the sequence of Noxo1. The destruction box (sequence RXXL where X represents any amino acid) is a degron found in many proteins [20]. Noxo1 has five destruction boxes, three in structured areas of the protein involved in the functional interactions of Noxo1 with its partners and two D-boxes in unstructured areas. We, therefore, mutated these two D-boxes in order to not interfere with the known functions of Noxo1 and to develop a form of Noxo1 less sensitive to degradation leading to an overactivation of Nox1 to develop a cellular model to allow high throughput screening for new Nox1 inhibitors.

## 2. Results

### 2.1. D-Box Mutations Mut1 of Noxo1 Induces a More Filamentous Pattern in Caco2 Cells

It has been described that Noxo1 is degraded by the proteasome [18,19]. Different degrons are known to be specific address markers of proteins to the proteasome, such as KEN-box, F-box, or D-box [21]. As shown in Figure 1, the Noxo1 sequence has five D-box (red lines). Three of the D-box are in structured domains (PX, orange, and SH3 domain, blue), while two others are present in unstructured parts of the protein (green). To address the role of the D-box of Noxo1 in its regulation, we mutated each of the two D-box present in the unstructured parts of the protein individually in order to not affect the interaction domains of Noxo1 with other partners of the NADPH oxidase complex. Indeed, mut1 was created by replacing the D-box amino acids with an alanine at amino acids 137 and 140 (RVIL to AVIA) and mut2 by replacing the D-box at amino acids 346 and 349 (RRAL to ARAA) (Figure 1). Noxo1 wt (NM_172167.3) and mutants were cloned in DDK or GFP vectors. After overexpression of these constructs in the colorectal cancer cell line Caco-2, we evaluated their expression using Western blot analysis. Using an anti-GFP antibody, we showed the expression of the wt Noxo1-GFP (lane 2), mut1 Noxo1-GFP (lane 3), or mut2 Noxo1-GFP (lane 4) proteins at the size of ~70 kDa in Caco-2 transfected cells. As expected, any detectable band of Noxo1 was observed in GFP-control transfected Caco-2 cells (lane 1) as a negative control, while the band of GFP was detected at a size of ~30 kDa (Figure 2A).

To examine the effects of each of these two mutations, we overexpressed in Caco-2 cells, GFP control vector, wt Noxo1-GFP, mut1 Noxo1-GFP, or mut2 Noxo1-GFP. Confocal analysis of the fluorescence image projection shown in Figure 2A revealed that while Caco2 cells overexpressing Noxo1 wt show a punctate pattern, Caco2 cells overexpressing mut1 or mut2 Noxo1 show a more filamentous pattern. The fact that mut1 shows the same phenotype as mut2 suggests that the observed phenotype is not related to the specific mutation position but seems to be more specifically related to D-box mutations. Since mut1 and mut2 constructs displayed similar Noxo1 filamentous pattern, we used the mut1 construct for the rest of the study. Detailed data image analysis was performed by defining four categories of objects based on the Noxo1 patterns from the confocal fluorescence images. These four categories are punctate, short sticks, long sticks, and filaments (Figure 2B, left panel). The result of the data analysis presented in Figure 2B (right panel) shows that wt Noxo1 and mut1 Noxo1 displayed the four categories but with different proportions. The wt Noxo1 cells exhibited a higher proportion of cells with the short stick category, while mut1 Noxo1 showed mainly cells with the filaments category. These results showed that mut1 induces a shift from short sticks pattern to a filaments pattern. To verify that the observed phenotype was not linked to an abnormal protein processing and folding in the endoplasmic reticulum (ER), we performed some co-staining of wt Noxo1 or mut1 Noxo1 with calreticulin and the protein disulfide isomerase (PDI), two known markers for ER. Immunofluorescence analysis did not show any abnormal localization of Noxo1 in the ER since we did not observe any colocalization of Noxo1 with calreticulin or PDI (Appendix A, Figure A1). This shows that the Noxo1 phenotype induced by mut1 is not due to a misfolding of the protein in the ER.

### 2.2. Noxo1 D-Box Mutation Increases ROS Production, Affects Mitochondrial Organization, and Increases Cytotoxicity in Colorectal Cancer Cell Lines

To test the functionality of mut1 Noxo1 compared to wt Noxo1, we measured the induction of superoxide production using lucigenin assay in Caco-2 and HT29-D4 cell lines. These two colorectal cell lines endogenously express all functional components of the Nox1-dependent NADPH oxidase complex [22]. Mut1 significantly increases superoxide production in both cell lines (Caco2: 140%, *p* < 0.05, Figure 3A; HT29-D4: 160%, *p* < 0.05, Figure 3B, ANOVA followed by Tukey’s test) compared to the overexpression of wt Noxo1 or ctl DDK plasmid. However, the overexpression of wt Noxo1 has no significant impact on superoxide production compared to control in Caco-2 cells and HT29-D4 cells (*p* > 0.05) (Figure 3). Mitochondrial organization (fusion-fission cycle) is known to be affected by oxidative stress [23].

During stress, the fusion of mitochondria allows functional mitochondria to complement dysfunctional mitochondria by diffusion and sharing of components between organelles [24]. We followed the organization of mitochondria in cells overexpressing wt Noxo1-DDK (Figure 4A, green) and mut1 Noxo1-DDK (Figure 4B, green) with the immunostaining of citrate synthase (Figure 4C,D, red), an enzyme expressed in the mitochondrial matrix [25]. Figure 4A–F shows that mut1 Noxo1 expression induces a reorganization of citrate synthase analyzed by immunofluorescence compared to that of wt Noxo1. Figure 4E shows the distribution of the mitochondrial puncta areas in wt Noxo1-DDK and mut1 Noxo1-DDK-expressing Caco-2 cells. Mut1 increases all but the smallest size of mitochondrial puncta as compared with those of wt Noxo1-DDK. Quantitative data showed that, in mut1 Noxo1-GFP-expressing Caco-2, the average mitochondrial puncta area was significantly bigger (7%) than that in wt Noxo1-DDK (*p* < 0.05, Student’s test) (Figure 4F). This increase in mitochondrial puncta area cannot be explained by an increase in citrate synthase expression, as shown by immunoblot in Figure 4G. Finally, overexpression of both wt and mut1 Noxo1 induces a significant increase in cytotoxicity compared to control vector overexpression. Wt Noxo1 increased the cytotoxicity by almost 36% (*p* < 0.05), while Mut1 increased cytotoxicity by 100% (*p* < 0.05, ANOVA followed by Tukey’s test) compared to ctl. This observation may be due to the greater increase in ROS production induced by mut1 Noxo1 compared to wt Noxo1.

### 2.3. Mut1 D-Box Mutation Affects Noxa1/Noxo1 Interaction and Leads to a Translocation from the Membrane Compartment to a Cytoskeletal Insoluble Fraction Compared to wt Noxo1 in Caco-2 Cell Line

To characterize the mechanism of action of the mut1 mutation, we observed the localization of mut1 Noxo1 and wt Noxo1 with the activator partner Noxa1 by fluorescence immunostaining on cells overexpressing the ctl vector, wt Noxo1 or mut1 Noxo1 in Caco-2 cell line. From the immunofluorescence images in Figure 5A, we see that the colocalization of wt Noxo1 (green) with endogenous Noxa1 (red) is rather at the level of the cell membrane involved in intercellular contacts, while much less for mut1 Noxo1 (green) with Noxa1 (red). The expression of mut1 Noxo1 seems to lead rather to colocalization with Noxa1, which is shared between membrane areas and areas of the presence of Noxo1 filament. Indeed, the quantitative analysis of colocalization Pearson coefficient (R colocalization) in Caco2 between Noxo1 and endogenous Noxa1 was significantly higher in wt noxo1 cells (3-fold, * *p* < 0.05) compared to that mut1 noxo1 (Figure 5B).

To identify whether Noxo1 is distributed in different cellular compartments that would have different behaviors depending on the lysis buffer, we used a cell fractionation kit and also performed cell lysis in buffers using different types of detergents.

Transfected Caco-2 cells with wt Noxo1 or mut1 Noxo1 were fractionated into cytosolic, membrane, and cytoskeleton fractions using a cell fractionation kit, and the same amounts (35 µg) of these fractions were analyzed by Western blot for specific markers: Glyceraldehyde-3-phosphate dehydrogenase (GAPDH) as cytosolic and membrane marker [26,27,28] and alpha Na,K-ATPase as a membrane marker [29]. GAPDH, and alpha Na,K-ATPase proteins were detected as a single band with molecular weights of 38 and 100 kDa, respectively, in transfected Caco-2 cells. In all transfected Caco-2 cells, GAPDH is mainly present in the cytosol and membrane fractions and at much lower levels in the cytoskeleton fraction (Figure 6A), consistent with its cytosol and membrane abundance. In contrast, alpha Na,K-ATPase was present in the membrane fraction at higher levels compared to the cytosol and cytoskeleton fractions, confirming its membrane enrichment. The results in Figure 6A show that wt Noxo1 is predominantly localized in the membrane fraction and, to a lesser extent, in the cytoskeleton fraction. This localization is reversed for mut1 Noxo1, which means it is mainly in the cytoskeleton fraction and, to a lesser extent, in the membrane fraction. Noxo1-wt and Noxo1-mut1 are not detected in the cytosolic fraction. Note that the addition of cytochalasin B (1 µM) for one hour before cell lysis does not alter the distribution of wt Noxo1 and mut1 Noxo1. This suggests that the actin cytoskeleton is not a direct player in Noxo1 compartmentalization. Immunostaining using an antibody against a membrane marker, alpha Na,K-ATPase, in cells expressing wt or mut1 Noxo1 presented in Figure 6B supports the observations obtained with the cell fractioning. Wt Noxo1 shows partial colocalization with the membrane alpha Na,K-ATPase mainly at the cell membrane, while this colocalization is not observed for the mut1 Noxo1. Finally, the immunoblot for the pellet and the supernatant of the cell lysis performed in different lysis buffers, using different detergents, shows that only the Laemmli buffer allows the extraction of the totality of the Noxo1 protein (wt or mut1) from the pellet and translocated it into the supernatant (Figure 6C). All other buffers (including RIPA) do not allow the extraction of the majority of Noxo1, which remains mostly in the pellet. Note that mut1 Noxo1 is found to a greater extent in this insoluble pellet fraction than the wt Noxo1, thus explaining the lower presence of mut1 Noxo1 in the soluble fraction (see Figure 4G and Figure 5A).

Overall, our results show that inactivation of the D-box at position 137–140 of Noxo1 leads to a transition of the protein from the membrane fraction to a cytoskeletal fraction, which seems insoluble to many detergents (NP40, n-octylglucoside, CHAPS, Brij20), and that only total lysis in Laemmli buffer allows a complete recovery of the Noxo1 protein in the Caco-2 cells. Independent of the mutation, the wt Noxo1 protein is also found but to a lesser extent in the insoluble fraction.

These results indicate that the D-box mutation mut1 in Noxo1 targets Noxo1 to an insoluble cytoskeletal fraction and decreases it in the soluble fraction, while it increases ROS production. This suggests that mut1-stimulated ROS production occurs in the insoluble or cytoskeletal fraction. We thus need to assess whether mut1-induced ROS production relies on Nox1-dependent NADPH oxidase complex. For this purpose, we used the HEK293 cell model to re-express the entire NADPH oxidase complex, as it is known that these cells don’t [22].

### 2.4. D-Box Mutation Mut1 in Noxo1 Increases ROS Production through Nox1-Dependant NADPH Oxidase

The HEK293 model is a classical model not expressing endogenously the Nox1-dependent NADPH oxidase complex in which all partners can be re-expressed. Expression of wt Noxo1 or mut1 Noxo1 and, Noxa1, Nox1 was performed in HEK293 cells. The result of the expression of these proteins analyzed by immunoblot is reported in Figure 7A. As shown in Figure 6C, to recover the whole Noxo1 protein, lysis was performed in Laemmli buffer. The analysis of superoxide anion production was evaluated by lucigenin luminescence and the results reported in Figure 7B show that ROS production is not stimulated compared to the control situation if the three partners are not expressed together (Noxo1/Noxa1/Nox1). When wt Noxo1 is expressed together with Noxa1 and Nox1, there is a significant (30%, *p* < 0.05; ANOVA followed by Tukey’s test) increase in ROS production compared to the ctl (lane 1, Figure 7B). Expression of mut1 Noxo1 with Noxa1 and Nox1 increases ROS production even more significantly (about 110%, *p* < 0.05; ANOVA followed by Tukey’s test) compared to the ctl (lane 1, Figure 7B). With the same ROS measurement method, this corresponds to the same order of magnitude for mut1 Noxo1 compared to control in terms of increase as those obtained on Caco-2 and HT29-D4 cells (Figure 3). The use of DPI (5 µM), an inhibitor of flavoproteins including Noxs, shows that the signal in the presence of Noxo1/Noxa1 and Nox1 is inhibited by more than 90%. However, it should be noted that in the ctl situation, DPI inhibits a part of the basal signal showing that other flavoproteins contribute to basal ROS production. These results confirm that the increase in ROS induced by mut1 Noxo1 is dependent on the NADPH oxidase Nox1 since in the absence of Nox1 expression and the activator Noxa1 mut1 Noxo1 is unable to induce ROS production.

We found it interesting to analyze the distribution of overexpressed Noxo1 and Noxa1 in soluble and insoluble fractions obtained in the HEK293 cell model following lysis in RIPA buffer (Figure 8). The results presented in this figure show that, as in Caco-2 cells, Noxo1 is mainly present in the insoluble fraction (pellet), and to a lesser extent, in the soluble fraction (supernatant). This result is the same whether Nox1 and Noxa1 are expressed endogenously or not, suggesting that it is not the assembly of the NADPH oxidase complex or its activity that impacts the localization compartment of Noxo1. In comparison, the Noxa1 protein is found mostly in the soluble fraction. GAPDH is also mainly present in the supernatant in all conditions, consistent with its natural localization, and used as loading ctl. These data agree with those observed in Caco-2 cells showing distinct localization for Noxo1.

Overall, our data show that the Nox1-dependent NADPH oxidase complex organizes as previously described at the cell membrane but that Noxo1 can leave the membrane compartment and reach the cytoskeleton compartment. This is accompanied in the cell by the clustering of Noxo1 as short filaments forming part of the cytoskeleton fraction. The mutation of D-box mut1 of Noxo1 amplifies this passage into the cytoskeletal compartment of Noxo1 and is accompanied by Noxo1 structures having long predominant filaments than with wt. Surprisingly mut1 Noxo1 induces an increase in NADPH oxidase activity, whereas the mutation favors the transfer from the membrane compartment where the active NADPH oxidase complex is assembled. The molecular mechanisms of how mut1 Noxo1 increases NADPH oxidase activity and why noxo1 is mainly enriched in the cytoskeleton compartment are still not well understood. Our initial idea of mutating the D-box to block the proteasomal degradation of Noxo1 to have a more stable protein remains to be tested in the context of these new results.

### 2.5. D-Box Mutation Mut1 in Noxo1 Does Not Stabilize Protein Turnover

In Caco-2 cells overexpressing wt Noxo1 or mut1 Noxo1, we studied the impact of MG132 (proteasome inhibitor used at 5–10 µM) on Noxo1 stability. Cyclin d1 expression was used as an endogenous degradation positive control and vinculin as a loading control. Since RIPA buffer do not extract all Noxo1 protein from the first pellet after lysis, the transfected Caco-2 cells were lysed in Laemmli buffer (Figure 9A). In Caco-2 cells overexpressing wt Noxo1, the results reported in Figure 9B (left panel) show that MG132 leads to a significant stabilization of cyclin d1 as soon as 12 h (9-fold), which is further enhanced after 24 h (12-fold) of treatment compared to corresponding ctl. In Caco-2 cells overexpressing mut1 Noxo1, MG132 also leads to a significant stabilization of cyclin d1 at both 12 h (2-fold) as well as 24 h (2,3-fold) of treatment (*p* < 0.05, ANOVA followed by Tukey’s test) compared to corresponding untreated 12 h ctl but at a lesser extent than for wt Noxo1. The expression of vinculin is not significantly affected by MG132 treatment at all time points in all conditions. Similar to vinculin, the expression of Noxo1 is also not significantly modified at all time points in all conditions (*p* > 0.05) (Figure 9B, right panel). These data show that in our experimental conditions, although cyclin d1 is stabilized by proteasome inhibition, this inhibition has no impact on Noxo1, whether it is wt or mut1. Similar results have been obtained following 1 and 4 h of MG132 treatment (Appendix A, Figure A4A). In addition, no ubiquitination was detected after immunoprecipitation of overexpressed wt Noxo1-DDK or mut1 Noxo1-DDK by anti-DDK antibody followed by Mass Spectrometry analysis on eight biological replicated experiments ((Appendix A, Figure A4B).

Overall, our results are not consistent with Noxo1 degradation by the proteasome, as shown by our data (Figure 9). We, therefore, tried to characterize the association of Noxo1 with the cytoskeleton.

### 2.6. D-Box Mutation Mut1 in Noxo1 Leads to Association of Noxo1 with Intermediate Filaments

As shown in Figure 6, wt Noxo1 is predominantly membrane-associated with partial localization to a cytoskeletal compartment resulting in cellular labeling as dots or short sticks. The mut1 Noxo1 switches the localization to a predominantly cytoskeletal localization and leads to filamentous cellular labeling. We, therefore, sought to characterize with which cytoskeletal element Noxo1 is associated. Figure 10 shows immunostaining of Caco-2 cells overexpressing wt Noxo1 or mut1 Noxo1 for Noxo1 and for the F-actin or microtubules. The F-actin organization was examined in Caco2 transfected cells using Alexa Fluor 594 phalloidin (red). WT Noxo1-DDK or mut1 Noxo1-DDK showed actin stress fibers crossing over the cytoplasm and bundles of F-actin beneath the plasma membrane. In addition, mut1 Noxo1-DDK cells did not display any morphological alterations compared with wt Noxo1-DDK transfected cells (Figure 10A, middle panels).

There is no particular colocalization of Noxo1 with F-actin except for a few peripheral membrane areas for either the wt or mut1 form of Noxo1 (Figure 10A). This rather uncommon colocalization is compatible with the regulation of Nox and actin by Rho-GTPases in membrane signaling hubs. Microtubule organization in cells transfected with wt Noxo1-DDK or mut1 Noxo1-DDK was assessed with alpha-tubulin antibody (Figure 10B, red). Wt Noxo1-DDK or mut1 Noxo1-DDK cells displayed microtubules in the cytoplasm as individual filaments until they were arrested by the cortical F-actin (Figure 10B, middle panels). The labeling of Noxo1 and microtubules show that there are some areas of colocalization between peripheral microtubules and Noxo1 (Figure 10B). Moreover, this peripheral punctiform colocalization seems to be compatible with the labeling of the ends plus microtubules classically involved in the regulation of focal contact and cell adhesion properties, where a role for Nox1 has been demonstrated [30]. These results, taken together, do not seem to show a difference between wt Noxo1 and mut1 Noxo1 with respect to their localization with F-actin or microtubules that could explain the shift of mut1 Noxo1 into a cytoskeletal compartment. Intermediate filaments represent the third type of cytoskeleton present in cells. Proteins such as keratins or vimentin form filaments of intermediate size to that of actin filaments and microtubules. Some publications refer to an interaction between Nox1 and keratin 18 [31] and some mass spectrometry experiments (Appendix A, Figure A4) performed on our models make us decide to analyze the association of Noxo1 with keratin 18. Of the three cytoskeletal components, keratin 18 shows the greatest difference in the labeling between wt and mut1 Noxo1 (Figure 11A). Keratin 18 organization in cells transfected with wt Noxo1-DDK or mut1 Noxo1-DDK was assessed with an anti-keratin 18 antibody (Figure 11A, red). Wt Noxo1-DDK or mut1 Noxo1-DDK cells displayed keratin 18 in the cytoplasm as individual filaments (Figure 11A, middle panels, red). Thus, keratin 18 organization is not different between wt Noxo1-DDK or mut1 Noxo1-DDK Caco-2 cells. The colocalization between wt Noxo1 and keratin 18 takes place in a punctate form at the crossing of the small wt Noxo1 rods, which seem to colocalize when they cross some keratin 18 filaments of the peripheral crown perpendicularly. Concerning mut1 Noxo1, the colocalization seems to be more intense than with wt Noxo1 and partially follows keratin 18 filaments (Figure 11A, right panels, yellow). Only part of the Noxo1 filaments is colocalized with keratin 18 filaments suggesting that Noxo1 may interact with another filament protein or form intracellular filaments itself. Figure 11B shows an immunoblot of Noxo1, keratin 18, Na,K-ATPase, and GAPDH expression in Caco-2 cells overexpressing wt Noxo1 or mut1 Noxo1 and lysed in RIPA buffer. Keratin 18 is well localized in the same insoluble compartment as Noxo1. There was no difference in keratin 18 expression between wt and mut1 Noxo1 cells in all conditions. These distributions are independent of whether F-actin is intact or depolymerized in all conditions.

Regarding vimentin distribution in Caco2 cells, confocal analysis of the fluorescence images shown in (Appendix A, Figure A2) revealed that while cells overexpressing wt Noxo1-DDK (green) displayed a punctate pattern for vimentin (red), which is found mainly in the cytoplasm and concentrated around the nucleus, Caco-2 cells overexpressing mut1 Noxo1-DDK (green) displayed a more filamentous pattern for vimentin (red, Figure A2) and distributed all over the cytoplasm. Thus, the mutation mut1 in D-box Noxo1 alters the vimentin pattern. Obvious colocalization at many spots or at filamentous sites for either the wt or mut1 form of Noxo1 (yellow). All our results show that the mutation of D-box 137–140 of Noxo1, in addition to its activating capacity of Nox1, leads to a localization of Noxo1 with intermediate filaments.

## 3. Discussion

Different studies reported proteasomal degradation for Noxo1 [18,19]. As ROS and NADPH oxidase production are involved in different pathologies [5,32], we try to develop a form of Noxo1 less sensitive to degradation leading to overactivation of Nox1 to develop a cellular model to allow high throughput screening for new Nox1 inhibitors. For this purpose, we mutated a putative degron in Noxo1 involved in degradation [32]. The Noxo1 sequence presents five D-box, with two of them in unstructured areas of the protein (D-box in 137–140 position and 346–349 position). We mutated each of these two D-box in Noxo1 individually. Confocal analysis revealed that cells overexpressing Noxo1 wt show a punctate pattern, while Caco-2 cells overexpressing mut1 or mut2 Noxo1 show a more filamentous pattern. The fact that mut1 shows the same phenotype as mut2 suggests that the observed phenotype is not related to the specific mutation position but seems to be more specifically related to D-box mutations. The mutation-induced an increase in ROS production measured by lucigenin assay in the three cell lines tested. For HT29-D4 and Caco-2 cells, we cannot exclude that part of the effect might result from other sources of ROS than Nox1 or from an impact on the antioxidant system. The reconstitution assay in HEK293 (overexpressing Noxo1, Noxa1, and Nox1) cells suggest that ROS is specific to Nox1 overactivation. This increase in ROS production is associated with mitochondrial reorganization and an increase in cytotoxicity, consistent with an increase in oxidative stress.

Although the model we have developed through mut1 mutation is responsible for an overactivation of Nox1 activity, it is not consistent with a lesser degradation of Noxo1 in Caco-2 cells since a potent, reversible, and cell-permeable proteasome inhibitor, MG132 has no effect on the level of Noxo1 expression. On the contrary, cyclin d1, used as a positive stabilization control [33], does show an increase in its expression level under the effect of MG132. It seems that the D-box mutation of Noxo1 leads to a part of the protein in a compartment associated with the cytoskeleton, which is difficult to solubilize by most of the lysis buffers except the Laemmli buffer. Noxo1 is usually thought to be a cytosolic subunit. This localization is mainly issued from overexpression experiments with Noxo1 immunofluorescence studies [34]. From our knowledge, there is no available data showing this localization using a cell fractioning assay. In our study, using cell fractioning, we showed that Noxo1-wt and Noxo1-mut1 are not detected in the cytosolic fraction but mainly in the membrane and cytoskeletal fractions. We cannot exclude that endogenous Noxo1 can behave differently.

The Noxo1 wt protein is predominantly present in the membrane fraction, while the mut1 Noxo1 protein is predominantly in the cytoskeletal fraction in colorectal cancer cells. This localization of mut1-Noxo1 in a cytoskeletal fraction corresponds to an increase in the insoluble fraction during cell lysis. Similar behavior has been observed in overexpression experiments in HEK293 cells. It is, therefore, plausible that the increase in Nox1 activity induced by mut1 Noxo1 occurs in the cytoskeletal-associated part. We also observe by immunofluorescence that Noxo1 does not associate with the actin cytoskeleton or microtubules but that there is an association with some intermediate filaments (keratin 18 and vimentin), which are indeed found in the insoluble part when using classical lysis buffers (RIPA). We thus show that Noxo1 associates in part in a cytoskeletal compartment and that mutation of D-box mut1 promotes this cytoskeletal localization. The significant overexpression of wt Noxo1 and mut1 Noxo1 shows that this compartment serves as an important reservoir of the protein and is, therefore, inconsistent with the rapid degradation of this stock. This effect is not specific to the cell model used (Caco-2 cells) since the absence of degradation by the proteasome is found in our experiments on HEK293 (Figure 9 and Appendix A, Figure A4). It should be noted that the absence of degradation of the Noxo1 protein in our experiments is not in agreement with the two published works on the subject [18,19]. It should be noted that neither paper shows direct evidence of Noxo1 ubiquitination. In the case of the Joo et al. study [19], there are several possible explanations for the discrepancy with our results. The method of cell lysis is not specified at any point [19]. Knowing that there are many immunoprecipitations presented in the results, it is likely that the lysis buffer used did not allow solubilizing of the whole pool of Noxo1. Moreover, the isoform of Noxo1 used in the study of Joo et al. is not specified either. It is, therefore, possible that the different isoforms of Noxo1 are not regulated in the same way. Initial work by Ueyama et al. [35] showed a very different distribution of the four isoforms (alpha, beta, gamma, delta) when expressed in HEK293 cells or COS7. Finally, it should be noted that the mut1 sequence that we mutated is not present in the murine homolog of Noxo1 when comparing human and mouse sequences in the NCBI protein database, which may also suggest a different regulation between species. Regarding the study of Haq et al., it is surprising that a deubiquitinase is responsible for an increase in Noxo1 ubiquitination [18]. Additionally, the interaction between CYLD deubiquitinase may well be indirect in the reported data. Again, in this study, the isoform is not specified, but the authors use RIPA buffer, which does not allow the extraction of the totality of Noxo1 from the cell lysate since most of the protein is eliminated after the first centrifugation in the insoluble part. It is likely from the data that there is an indirect interaction of CYLD and Noxo1, showing some correlation in the data without causal relationships being established. It is interesting to note that Joo et al. [19] show high expression of Noxo1 in 159 of 222 colorectal cancer patients. The authors link this to the stabilization of Noxo1 in cancer when it may simply be an increase in expression or a very different mechanism. Interestingly, the colocalization that we have demonstrated with certain intermediate filaments, including keratin 18 (CK18), would provide an attractive alternative explanation. CK18 is with keratin 8, a proven marker of colorectal cancer [36]. CK18 is ubiquitinated and degraded by the proteasome, and Nox1 has been reported to block this degradation. CK18 accumulation induced by Nox1 is consistent with the persistence of fetal-type CK18 protein in many epithelial carcinomas [36]. These findings do not call into question the conclusions on the role of Noxo1 in cancer progression proposed by Joo et al. [19] and Haq et al. [18] but only the proposed mechanism linking Noxo1 to the ubiquitin ligase Cbl or the deubiquitinase CYLD, which would be rather indirect. Indeed, whether in these two studies or in the present paper, there is no experimental result allowing to affirm any direct degradation of Noxo1 by the proteasome. Such degradation may exist but may be difficult to demonstrate if it occurs only in the membranal or cytosolic compartment, and the protein can escape this degradation by going into some cytoskeletal associated compartment.

Concerning the role of D-box mut1 in the sequestration of Noxo1 in the intermediate filaments compartment, the mechanism remains to be elucidated. One possibility is that the association of Noxo1 with the cytoskeleton is to compartmentalize ROS production, as suggested for Nox4 [37]. This mechanism seems to be able to enlighten us on an important carcinogenesis mechanism in colorectal tissues linking proteasome, the ubiquitin ligase Cbl, the deubiquitinase CYLD, keratin 18 and Noxo1 in a common signaling hub. Thus, the present study provides the first evidence that the D-box of Noxo1 modulates the normal membrane-cytoskeleton balance of Noxo1.

## 4. Materials and Methods

### 4.1. Materials and Reagents

The sequences for Noxo1 wt (NM_172167.3), mut1, mut2 inserted in PCMV6-beta-Noxo1-C-DDK and pcDNA3.1-NeGFP-beta-Noxo1 as well as control vector were obtained from Origene Technologies (Rockville, MD, USA). We choose to work on the beta-Noxo1 isoform since it’s the form being expressed in colonic and hepatic cells. Gamma-Noxo1 being express in testis and alpha and delta never reported in tissues [37]. Fetal bovine serum (FBS, Lonza, Basel, Switzerland), trypsin-EDTA 0.5%, Dulbecco modified Eagle medium (DMEM), sodium pyruvate, and DAPI (D1306) were obtained from Gibco-BRL (Invitrogen Corporation, Inchinnan, Scotland-UK). The following reagents were used: apocynin (Sigma A10809-25, Saint-Quentin-Fallavier, France), an inhibitor of ROS production, stock at 100 mM and used at 0.5 mM (validated by previous work); diphenyleneiodonium chloride DPI (CAS 4673-26-1, Merck Millipore, Darmstadt, Germany) inhibitor of ROS production, stock at 10 mM and used at 10 μM (concentration validated by previous work); ML171 (cat 492,002 Calbiochem, Darmstadt, Germany), a specific inhibitor of Nox1, stock at 10 mM and used at 1.5 µM (in accordance with the IC50 values provided by the manufacturer); cytochalasin B (C-6762 Sigma, France), stock at 10 µg/µL and used at 10 µg/mL; latrunculine B (ref 428,020 Medchmexpress, Sollentuna, Sweden), stock at 1 mg/mL and used at 1 µg/mL; taxol (Enzo-BML-T104-0005, Lausen, Swizerland), stock at 5 mg/mL and used at 5 µg/mL; sodium orthovanadate (Sigma T7765, Saint-Quentin-Fallavier, France), stock at 1 mM and used at 1 mM; Nocodazole (Sigma M1404), stock at 10 mg/mL and used at 10 µg/mL; MG 132 (Sigma-c2211) stock at 5 mM used at 5 µM or 10. Collagen type 1 (C3867-1VL Sigma, Saint-Quentin-Fallavier, France) 4 mg/mL used at 10 µg/mL. Other chemicals were purchased from Sigma-Aldrich (St. Louis, MO, USA), NADPH 11630-50MG Sigma, and lucigenin (M8010-1g Sigma). The ProteoExtract Subcellular Proteome Extraction Kit 539790 (Millipore, Darmstadt, Germany). A/G ultralink resin beads (53132 Thermo Fisher Scientific, Waltham, MA, USA). For all the antibodies we used, see Table 1.

### 4.2. Cell Culture and Transfections

Two human colon carcinoma cell lines, HT29-D4 and Caco-2, and HEK 293 cells (derived from human embryonic kidney) were grown in DMEM supplemented by 10% FBS, 25 mM D-Glucose, sodium pyruvate (1% *v*/*v*), 1% non-essential amino acids and 1% penicillin/streptomycin at 37 °C in a humidified atmosphere with 5% CO_2_. The HT29-D4 cells were grown at a maximum confluence of 80%. Then the cells were collected after trypsin-EDTA treatment and transfected by Amaxa nucleofector according to the manufacturer’s protocol (KIT V, VCA-1003 Lonza, Basel, Switzerland). Then, 2 × 10^6^ HT29-D4 cells were centrifuged at 200× *g* for 5 min resuspended in 100 µL transfection buffer containing 2 µg of the appropriate plasmid, and then transferred into the electroporation cuvette. Program T020 was used for HT29-D4 cell line electroporation. The adhering cells were seeded on 6-well plastic plates or on collagen type 1-treated glass coverslips at a concentration of 10 µg/mL and incubated at 4 °C overnight or at 37 °C for 4 h. After 24 h of seeding (15,000 cells/well in 96-well plates, 40,000 cells/well in 24-well plates, and 80,000 cells/well in 12-well plates or 500,000 cells/well in 6-well plates), cells reached 80% confluence before being transfected or treated. Caco-2 and HEK293 cells were transfected by lipofection according to the Invitrogen protocol and with calcium chloride, respectively, for 12–48 h with constructs encoding Nox1, Noxo1, and Noxa1.

### 4.3. Subcellular Fractionation, Cell Extracts, and Immunoblot Analysis

Caco-2 cells transfected with wild and mutated Noxo1 coupled to DDK or GFP were fractioned into cytosolic, membrane, and cytoskeleton fractions using the ProteoExtract Subcellular Proteosome Extraction Kit according to the manufacturer’s instructions. The purity of the fractions was assessed by Western blotting with specific markers. All fractions were separated into aliquots, and protein concentrations were determined by the bicinchonic acid test (BCA) (Dallas, TX, USA) according to the manufacturer’s instructions. After 5 min boiling, Laemmli buffer with 6% 2-mercaptoethanol was added to aliquots containing 30–70 µg protein, which were loaded per lane and analyzed on 10% sodium dodecyl sulfate (SDS)-polyacrylamide gel electrophoresis (PAGE) using a MiniBlot system (Bio-Rad, Marnes-La-Coquette, France). After electrophoresis, the proteins were transferred onto Hybond-ECL nitrocellulose membranes (Amersham Biosciences, Buckinghamshire, UK) in transfer buffer (25 mM Tris, 192 mM glycine, 20% ethanol). Before blocking, the blots were stained with Ponceau red to visualize transfer efficiency. The blots were blocked in Tris-buffered saline (TBS and 0.05% Tween20, TBS-T) containing 5% milk for 1 h at RT and sequentially incubated overnight at 4 °C in TBS-T containing 5% milk and the following primary antibodies: mouse monoclonal GAPDH, Na,K-ATPase, DDK antibodies see Table 1. After incubation with primary antibodies, blots were washed 3 times for 5 min in TBS-T, incubated with corresponding horseradish peroxidase (HRP) secondary antibodies, all from cell signaling (Table 1): anti-mouse IgG-HRP was used to reveal GAPDH and Na,K-ATPase, and goat anti-rabbit IgG Fc-HRP to reveal DDK, diluted in TBS-T containing 5% milk for 1 h at RT and washed 3 times for 5 min each with TBS-T. Finally, proteins were detected using the chemiluminescence HRP substrate (Merck Millipore) and visualized using the chemiluminescence imaging system G-Box (Syngene, Cambridge, UK).

Since Noxo1 has been shown to be present in different cellular compartments, different lysis buffers were used to identify which buffer solubilizes this protein present in the pellet. The composition of buffers used are as follows: RIPA buffer (50 mM Tris-HCl, pH 8, 150 mM NaCl, 1% NP40, 0.5 mM EDTA, 1% inhibitors of protease and phosphatase); the derivative of RIPA buffer (4.3 mL TBS 1X, 50 µL SDS 10%, 10 µL EDTA 0.5 M, 500 µL triton 10× and 1% inhibitors of protease and phosphatase); CHAPS buffer (40 mM HEPES, PH7.4, 120 mM NaCl, 2 mM EDTA, 10 mM CHAPS, inhibitors of protease and phosphatase); BRIJ buffer (50 mM HEPES, 100 mM NaCl, 0.01% BRIJ-20, 1 mM MnCl2, 2 mM DTT, 1% inhibitors of protease and phosphatase); N-octylglucoside buffer (1.5% octyl glucoside, 20 mM HEPES-NaOH, PH 7.4, 1 mM EDTA, 150 mM NaCl, 5 mM MgSO4, 1% inhibitors of protease and phosphatase). Laemmli buffer (250 µL Tris-HCl 1 M, pH 6.8, 1.25 mL SDS 20%, 500 µL glycerol, 40 µL EDTA, 1% inhibitors of protease and phosphatase). After 30 min of lysing the cells with 120 µL of lysis buffer per well of 6 well-plates at 4 °C, lysates were transferred to Eppendorf tubes and centrifuged for 10 min at 14,000 rpm at 4 °C. The protein concentration of the obtained supernatant was determined. After electrophoresis, the proteins were transferred onto Hybond-ECL nitrocellulose membranes in transfer buffer. Before blocking, the blots were stained with Ponceau red to visualize transfer efficiency. The blots were blocked in Tris-buffered saline (TBS and 0.05% Tween20, TBS-T) containing 5% milk for 1 h at RT and sequentially incubated overnight at 4 °C in TBS-T containing 5% milk and the appropriate primary antibodies. Finally, proteins were detected using the ECL Prime Chemiluminescence Kit (Roche Diagnostics, Meylan, France) and visualized using a chemiluminescence imaging system (UVITEC, Cambridge, UK). The band intensity was quantified using the NIH ImageJ software version 1.53q [38].

### 4.4. Measurement of Superoxide Production

After 48 h of transfection with ctl DDK, wt, and mut1 Noxo1 DDK, cells were trypsinized and counted. For the analysis of superoxide production, 200,000 cells from each condition were incubated in DMEM without phenol red with 1 mM of NADPH (cofactor of NADPH oxidases) and 30 μM of lucigenin. For each condition, the obtained mix was used to seed 3 wells (50,000 cells/well) on a 96-well white plate. Superoxide production was calculated by integrating the luminescence values measured every minute for a period of 45 min at 37 °C using a Fluoroskan plate reader (FL Fluoroskan Ascent, Labsystems, MA, USA). In order to assess the cell viability after superoxide measurements, the cells were fixed with glutaraldehyde (1%) for 10 min and stained with violet crystal (0.1%) for 30 min. After several washes with PBS, the cells were lysed in 1% SDS, and the optical densities were measured using a plate reader (Multiskan RC, Labsystems), reflecting the number of cells. Results obtained with lucigenin (RLU) were normalized using the OD values of the purple crystal for each condition. They were then compared to the control condition and expressed as a percentage. In some sets of experiments, we used diphenyleneiodonium chloride (DPI), a well-known NADPH oxidase inhibitor, to inhibit superoxide anion production.

### 4.5. Immunofluorescence

After 24 h of transfection with ctl DDK/GFP, Noxo1 wt, or mut1 DDK (GFP), the cells were fixed with methanol at −20 °C or paraformaldehyde (PFA) 4% at RT for 20 min followed by 3 times rinses with PB 0.12 M. In the case of PFA fixation; the cells were permeabilized in a solution containing 0.1% Triton X-100 diluted in PB 0.12 M for 5 min at RT. After blocking in a solution containing 3% BSA diluted in PB 0.12 M for 30 min at RT, the cells were incubated with primary antibodies overnight at 4 °C. The next day, the cells were washed 3 times in PB 0.12 M under agitation and incubated with appropriate secondary antibodies diluted in the blocking solution for 1 h at RT in the dark. Nuclei were counterstained with 5 μg/mL DAPI for 30 min at RT. Cells were rapidly rinsed 3 times in distilled water and let to dry before mounting on Superfrost glass slides using a long-lasting anti-fade medium (P36394, Thermo Fisher Scientific, MA, USA) and stored at −20 °C until used. Regarding HCS experiments, the plates (Microplaque CellCarrier Ultra 96 well) were stored in PB 0.12 M containing azide (0.05%) at 4 °C. Labeling specificity was assessed under the same conditions by incubating some coverslips/wells in a solution omitting the primary antibodies. In all cases, no staining was detected. The images were acquired using a confocal laser-scanning microscope (LSM 700, Zeiss, Jena, Germany) with a x40 oil objective, and analysis of immunostaining images was performed using ZEN software (Zeiss, Paris, France) or acquired using the PerkinElmer HCS device (PerkinElmer, Villebon-sur-Yvette, France) with a x40 water objective and analysis of immunostaining images was performed using Harmony software (PerkinElmer, Villebon-sur-Yvette, France). In some sets of experiments, the analysis of the colocalized fluorescence intensity on Caco2 double-immunolabeled Noxa1/Noxo1 cells was performed by measuring the R colocalization using the Pearson coefficient, with ImageJ2 version 2.3.0/1.53 q. software and “coloc2” plug-in.

### 4.6. Cytotoxicity Test and Blue Trypan Exclusion Test

The goal of the exclusion test is to determine the number of viable and dead cells in a cell suspension. The cells by excluding or absorbing the dye, we can visually determine which dead cells are marked in blue from living cells. After 48 h of transfection, the cells were trypsinized, and an aliquot of 10 μL of the suspended cells was mixed with 10 μL of trypan blue (0.4%). We took 10 µL of the mix that we deposited on the Malassez counting chamber and counted the colorless viable and the dead blue cells under an optical microscope. The percentage of cellular mortality is calculated as follows: Cell mortality = (total number of dead cells/total number of cells) × 100.

### 4.7. Data and Statistical Analysis

For the measurement of mitochondrial surface, a macro was developed in ImageJ software [38] to make a segmentation delimiting first the contours of the cells using Noxo1 DDK immunolabeling. This later step was followed by cell thresholding, allowing keeping intensities greater than 3 pixels. This allows segmenting of the citrate synthase spots displaying a surface greater than 0.1 µm^2^. The citrate synthase spots correspond to the mitochondrial surface (Appendix A, Figure A3). Wilcoxon Mann–Whitney test was used to analyze whether there is any statistical difference between the mitochondrial surface in Noxo1 wt and mut1 DDK. For the ROS measurement assay, one-way ANOVA followed by post hoc Tukey’s test was used to compare means of ROS production between different groups.

### 4.8. High Content Screening (HCS) Data Analysis

Quantification of the number of cells that express the different phenotypes of Noxo1 GFP after HCS acquisition was performed using the Harmony software (PerkinElmer, Villebon-sur-Yvette, France) associated with the HCS Operetta from Perkin-Elmer. Using the texture algorithm allowed us to identify four phenotypes in the first set of experiments. The software allowed us to analyze the different fields and compare the acquired images with the previously introduced reference images to quantify the percentage of cells with each phenotype (machine learning). Data were obtained on a set of three independent experiments with a total of 1794 cells quantified. The Chi^2^ test application shows statistically significant differences between wt and mut1 Noxo1.

### 4.9. Statistical Analysis

All experiments were performed at least 3 times with different culture series or independent cultures. Student’s *t*-test was used to compare 2 groups. ANOVA analysis, followed by Tukey’s post hoc test, was used for multiple comparisons. All data were expressed as the mean ± SEM. Statistical significance was set at * *p* < 0.05.

## Figures and Tables

**Figure 1 ijms-24-04663-f001:**
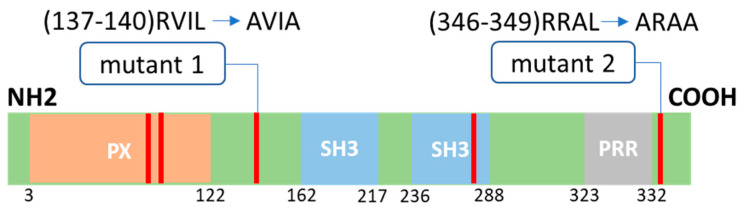
Noxo1 structure and D-box mutations. Noxo1 sequence contains four functional domains, including one phosphoinositide-binding structural domain (PX, orange), two Src homology 3 domains (SH3, blue), one proline-rich region (PRR, grey), and five putative D-box motifs (RXXL, red lines). We generated two mutants of Noxo1 as indicated, mutant 1 (mut1) and mutant 2 (mut2), with the mutation of the two D-box present in the unstructured part of the protein (green).

**Figure 2 ijms-24-04663-f002:**
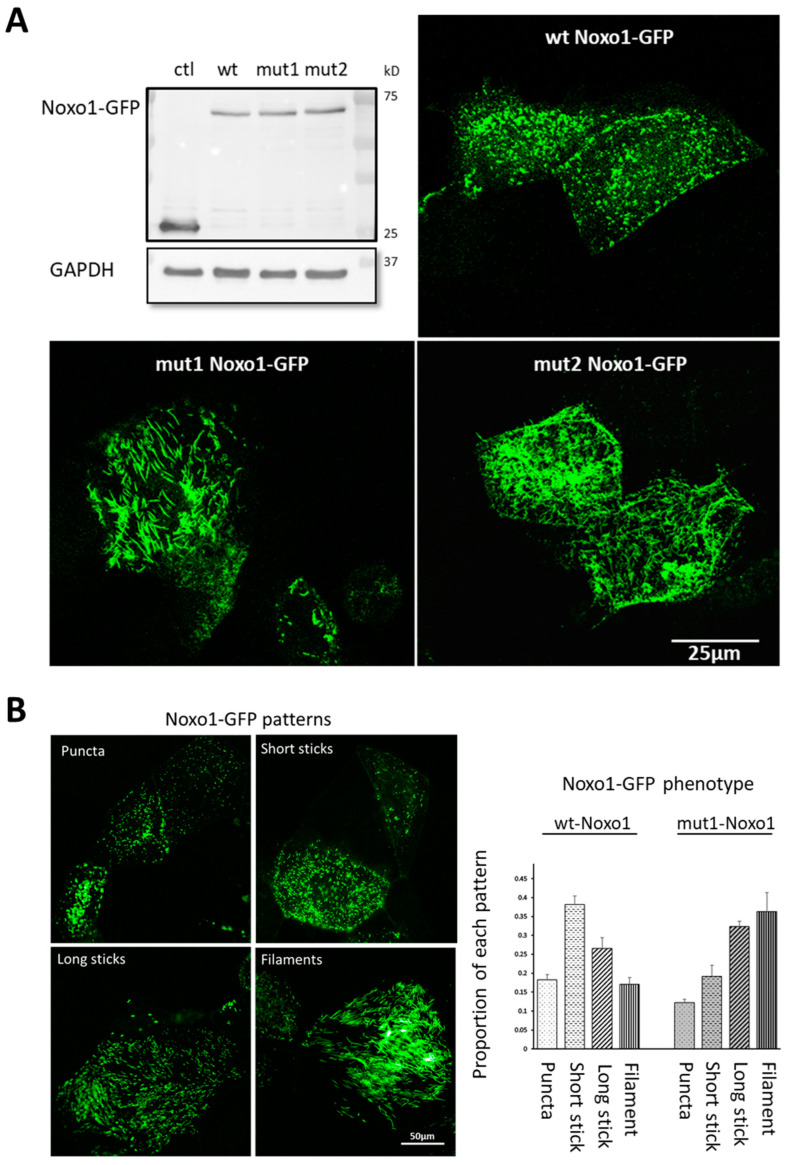
Effects of the overexpression of Noxo1-GFP on the pattern of Noxo1 in Caco-2 cells. (**A**, **upper left**) Immunoblot of control GFP (ctl), wt Noxo1-GFP, mut1-GFP, and mut2-GFP overexpression. GAPDH was used as the loading control. The ctl corresponds to transfection with the GFP vector. (**A**, **upper right** and **lower panels**) Immunofluorescence images for GFP overexpressing wt Noxo1-GFP, mut1 Noxo1-GFP, or mut2 Noxo1-GFP, respectively. Confocal immunofluorescence imaging projections show that the wt Noxo1-GFP displays a punctate pattern while mutants exhibited a more filamentous pattern. In all panels, scale bar: 25 µm. (**B**, **left**) illustrate different Noxo1-GFP patterns, including puncta, short sticks, long sticks, and filaments. Scale bar in all panels: 50 µm. (**B**, **right**) quantitative analysis of each pattern in both wt Noxo1-GFP and mut1 Noxo1-GFP obtained using HCS Operatta. Scale bar: 50 µm.

**Figure 3 ijms-24-04663-f003:**
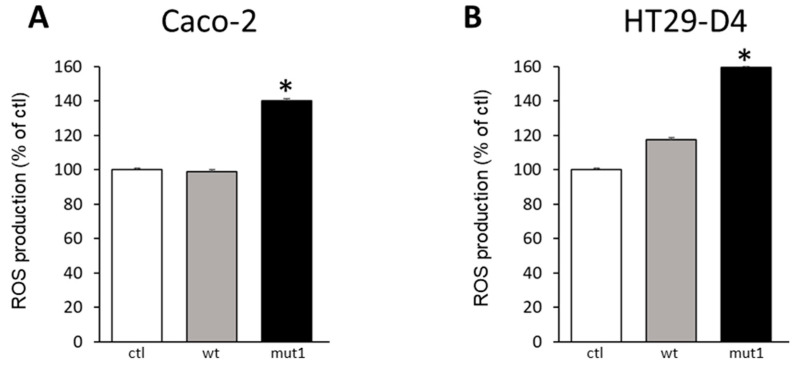
ROS production in Caco-2 and HT29-D4 cells overexpressing ctl, wt Noxo1-DDK or mut1 Noxo1-DDK. (**A**,**B**) Illustrate ROS production in Caco-2 and HT29-D4 cells overexpressing ctl, wt Noxo1-DDK, or mut1 Noxo1-DDK. Mut1 Noxo1 overexpression induced a significant increase in ROS production in both cell types compared to ctl or WT. * *p* < 0.05 compared to ctl.

**Figure 4 ijms-24-04663-f004:**
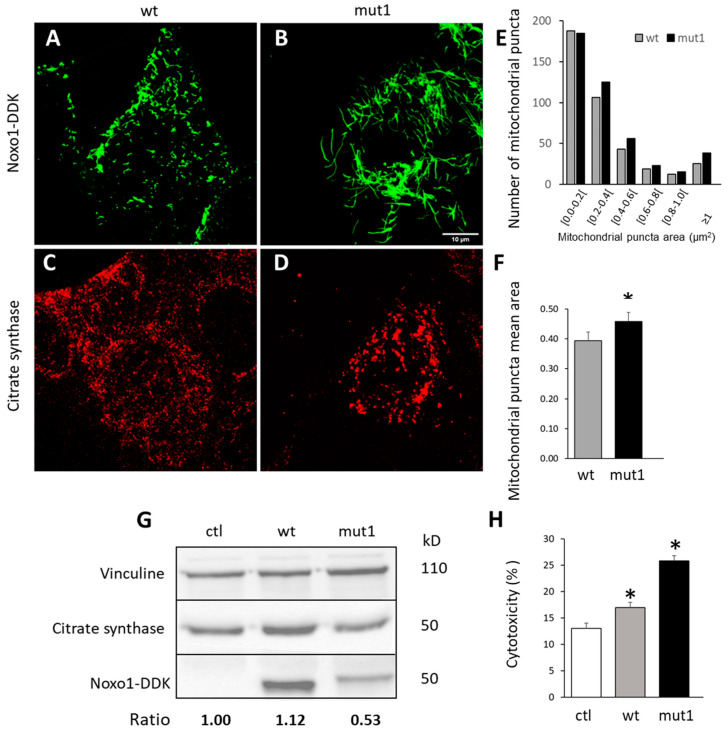
Mut1 Noxo1-DDK overexpression affects the mitochondrial organization and cellular toxicity in Caco-2 cells. Mitochondria were revealed using antibodies directed against the specific mitochondrial enzyme citrate synthase (**A**,**C**) in wt and mut1 Noxo1-DDK expressing cells (**B**,**D**). Scale bar in all panels: 10 µm. (**E**,**F**) Quantification of the apparent mitochondrial areas were measured in wt and mut1 Noxo1-DDK expressing cells. Histograms showing the distributions of the mitochondrial puncta area classes (**E**) and the average mitochondrial puncta area in wt and mut1 Noxo1-DDK expressing cells (**F**). (**G**) Immunoblot of the total expression of citrate synthase, Noxo1, and vinculin in wt and mut1 Noxo1-DDK expressing cells. The lower numbers referred to the ratio of citrate synthase/vinculin. (**H**) Cellular toxicity assessed using blue trypan vital staining followed by cell counting in ctl, wt, and mut1 Noxo1-DDK transfected cells. Altogether, mut1 Noxo1 overexpression induced mitochondrial aggregation and cellular toxicity compared to wt Noxo1. * *p* < 0.05 compared to ctl.

**Figure 5 ijms-24-04663-f005:**
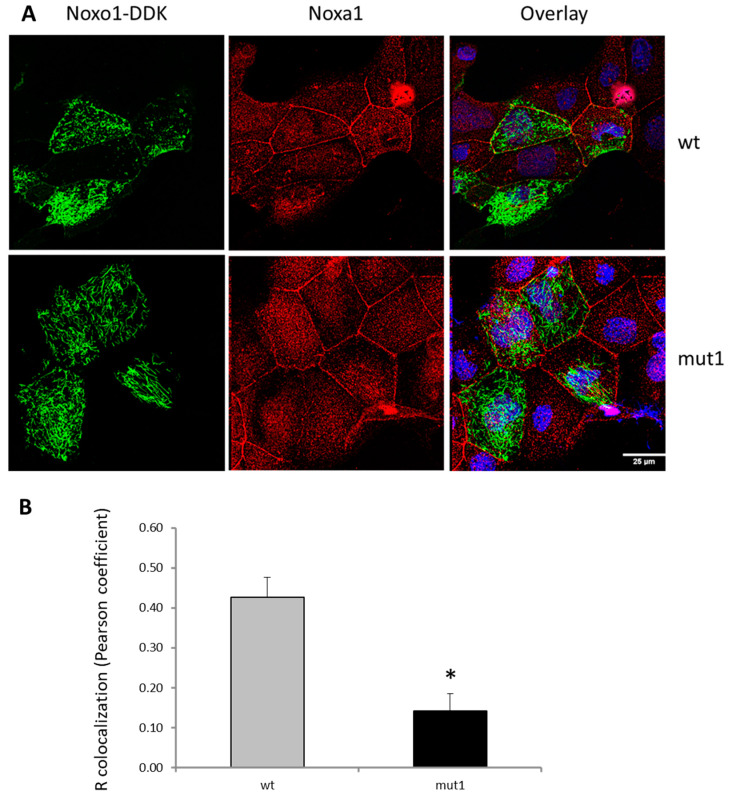
Effect of mut1 D-Box mutation on the Noxa1/Noxo1 interaction in CaCo-2 cells. (**A**) Fluorescence image of endogenous Noxa1 protein and overexpressed Noxo1-DDK. Zeiss 40x oil Zoom1. Both WT Noxo1-DDK and mut1 Noxo1-DDK did not alter the membrane localization of Noxa1. Scale bar in all panels: 25 µm. (**B**) Quantification of colocalization (R colocalization, Pearson coefficient) of fluorescence signal from Noxo1 (wt or mut1) and Noxa1 in Caco2 cells (N = 21 ROI), * *p* < 0.05).

**Figure 6 ijms-24-04663-f006:**
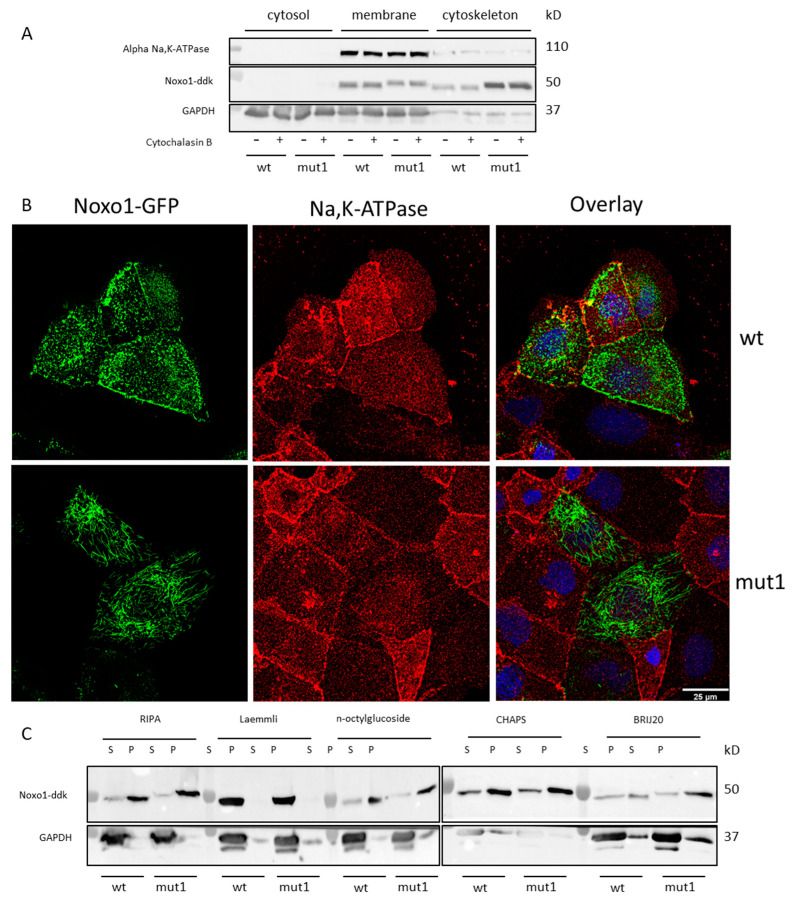
Wt and mut1 Noxo1 localization after subcellular fractioning. (**A**) Subcellular fractioning of Caco-2 cells were transfected by wt or mut1 Noxo1-DDK was performed using ProteoExtraction Subcellular Proteosome Extraction Kit according to manufacturer recommendations. The different fractions were revealed for Noxo1-DDK, alpha Na,K-ATPase (control for membrane), and GAPDH (control for cytosol and membrane) by immunoblot. Mut1 Noxo1 is predominantly present in the cytoskeletal fraction, while wt Noxo1 is predominantly present in the membrane fraction. F-actin depolymerizing agent, cytochalasin B (1 µM, 1 h before lysis) didn’t affect wt and Mut1 Noxo1 subcellular localization. (**B**) Double immunofluorescence of alpha Na,K ATPase protein (red), and Noxo1-GFP (green). Scale bar in all panels: 25 µm. (**C**) Caco-2 cells were transfected by wt or mut1 Noxo1-DDK and lysed in different buffers (RIPA; Laemmli, n-octylglucoside; CHAPS, Brij20) followed by separation of supernatant (S) and pellet (P) after centrifugation (15,000× *g*, 5 min). Noxo1-DDK and GAPDH were revealed by immunoblot using DDK and GAPDH antibodies.

**Figure 7 ijms-24-04663-f007:**
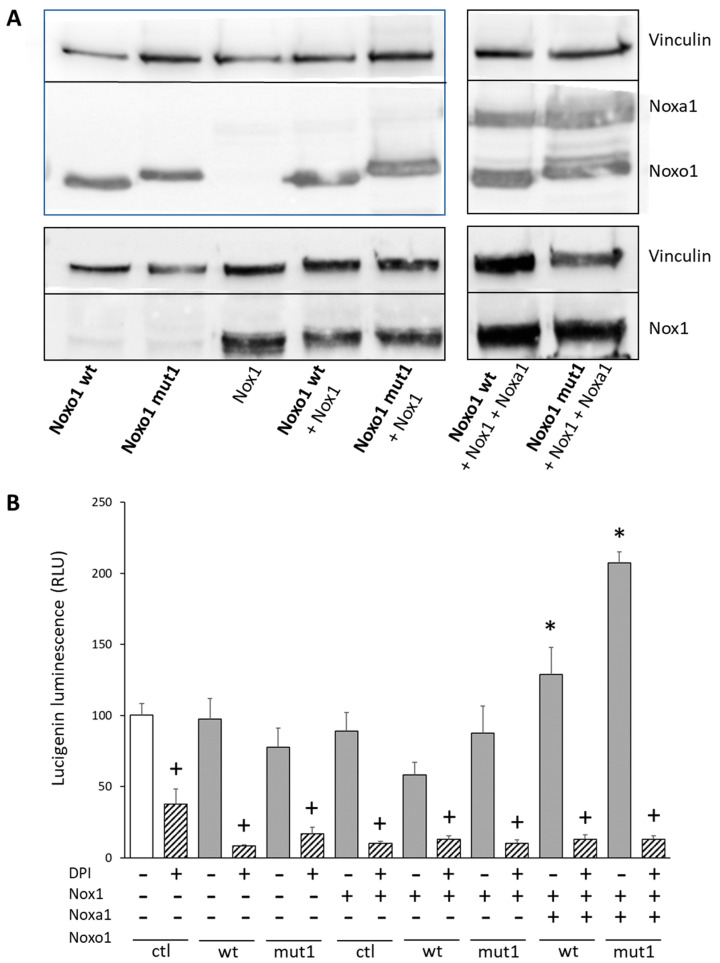
Mut1 Noxo1 increases ROS production in HEK293 cells expressing Nox1-dependent NADPH oxidase complex. (**A**) Immunoblot of Nox1-dependent NADPH oxidase complex reconstitution. Nox1 was revealed by an anti-Nox1 antibody, Noxo1 and Noxa1 were revealed using an anti-DDK antibody. Vinculin was used as the loading control. Lysis was performed in Laemmli buffer. (**B**) ROS production by HEK293 cells transfected with pCMV6 entry vector (ctl), wt Noxo1-DDK or mut1 Noxo1-DDK along with Noxa1 and Nox1 measured by lucigenin luminescence. Results represented mean +/− S.E.M percentage of ctl (lane 1) issued from four independent experiments realized in triplicate. In some assay DPI (10 µM, 45 min), a flavoprotein inhibitor known to inhibit Nox1 was used. Both wt Noxo1 or mut1 Noxo1 in the presence of Nox1 and Noxa1 increase ROS production. Mut1 Noxo1 is more potent than wt Noxo1 in inducing ROS production. * *p* < 0.05 compared to ctl (lane1); + *p* < 0.05 compared without inhibitor.

**Figure 8 ijms-24-04663-f008:**
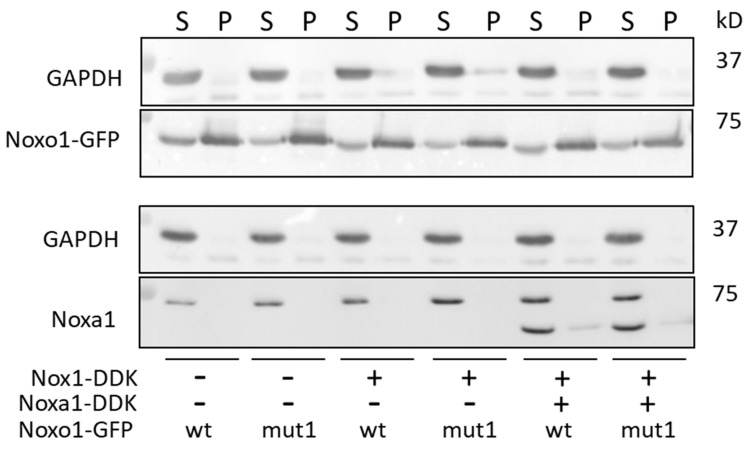
Effect of the expression of NADPH oxidase complex on the distribution of Noxo1-GFP in soluble and insoluble HEK293 cells fractions lysed in RIPA buffer. HEK 293 cells were transfected with wt Noxo1 or mut1 Noxo1 and, Noxa1, Nox1. The result of the expression of these proteins was analyzed by immunoblot using an anti-Noxa1 antibody for Noxa1, an anti-GFP antibody for Noxo1, and an anti-GAPDH antibody for GAPDH. Both wt Noxo1-GFP or mut1 Noxo1-GFP are mainly present in the insoluble fraction (pellet), and to a lesser extent, in the soluble fraction (supernatant). This distribution is independent of whether Nox1 and Noxa1 are expressed endogenously or not. Noxa1 protein is observed almost exclusively in the soluble fraction of RIPA buffer. GAPDH is also mainly present in the supernatant in all conditions and used as the loading ctl.

**Figure 9 ijms-24-04663-f009:**
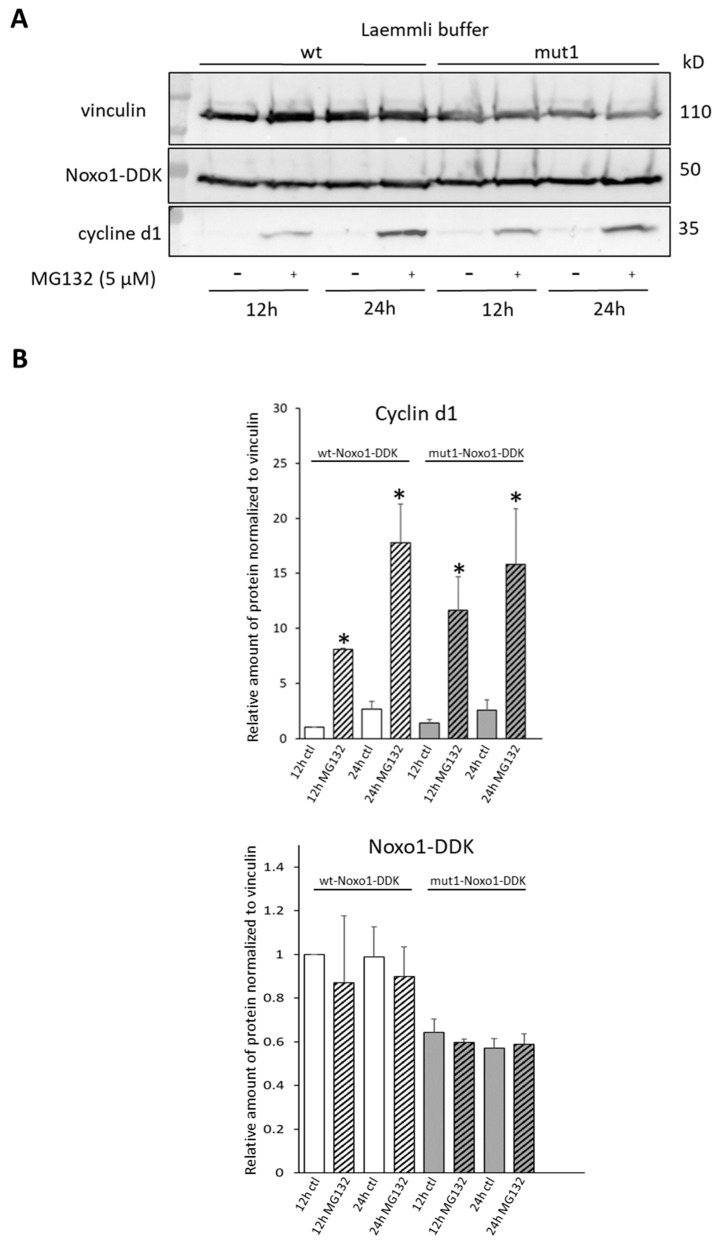
Noxo1-DDK stability analysis using proteasomal inhibitor MG132 in Caco-2 cells overexpressing wt or mut1 Noxo1-DDK and lysed in Laemmli buffer. (**A**) Caco-2 cells were treated with MG132 (for 12 or 24 h, 5 µM). Immunoblot of vinculin, Noxo1-DDK, and cyclin d1. Vinculin was used as the loading control, and cyclin d1 was used as the positive control for proteasomal degradation. (**B**) Quantification of immunoblots for cyclin (upper panel) and Noxo1-DDK (bottom panel). Results represented mean +/− S.E.M normalized to ctl vinculin using the untreated 12 h time point as reference. * *p* < 0.05 compared without MG132.

**Figure 10 ijms-24-04663-f010:**
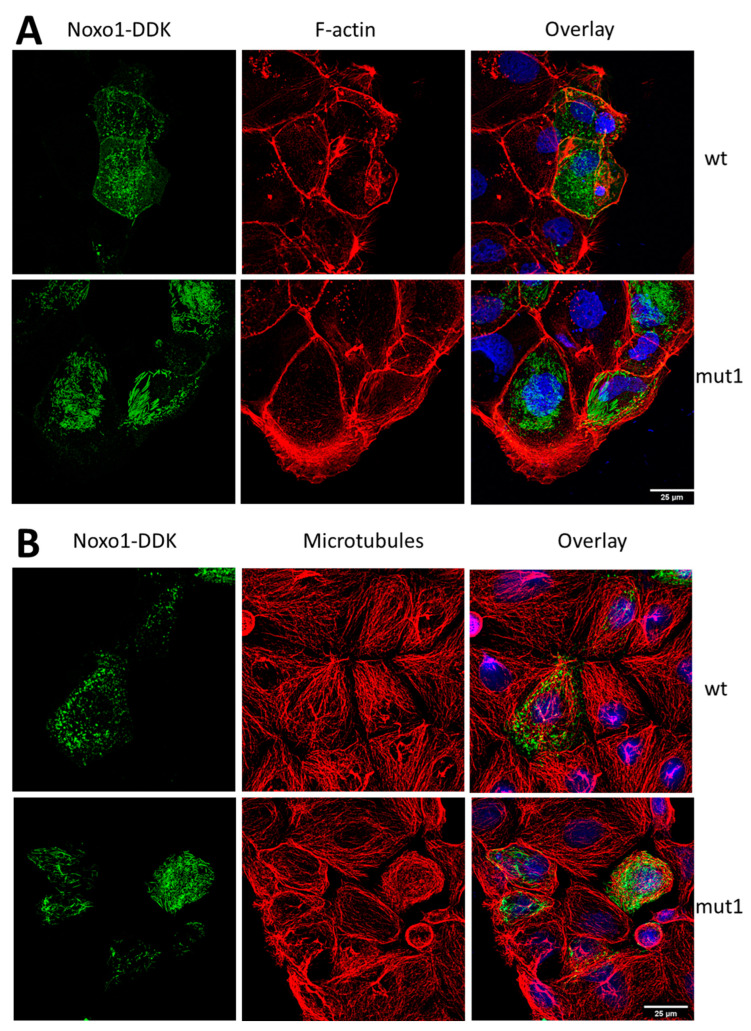
Immunofluorescence Noxo1 with F-actin or microtubules in Caco-2 transfected with wt Noxo1- or mut1 Noxo1-DDK. (**A**) Immunofluorescence of Noxo1-DDK and F-actin. Wt-Noxo1 and mut1-Noxo1 (green, upper and lower left panel), F-actin (red, middle panel), and overlay corresponding to Noxo1-DDK/F-actin/nucleus (using DAPI) (left panel). (**B**) Immunofluorescence of Noxo1-DDK and microtubules. Wt-Noxo1 and mut1-Noxo1 (green, upper and lower left panel), microtubules (red, middle panel), and overlay corresponding to Noxo1-DDK/microtubules /nucleus (using DAPI) (left panel). Scale bar in all panels: 25 µm.

**Figure 11 ijms-24-04663-f011:**
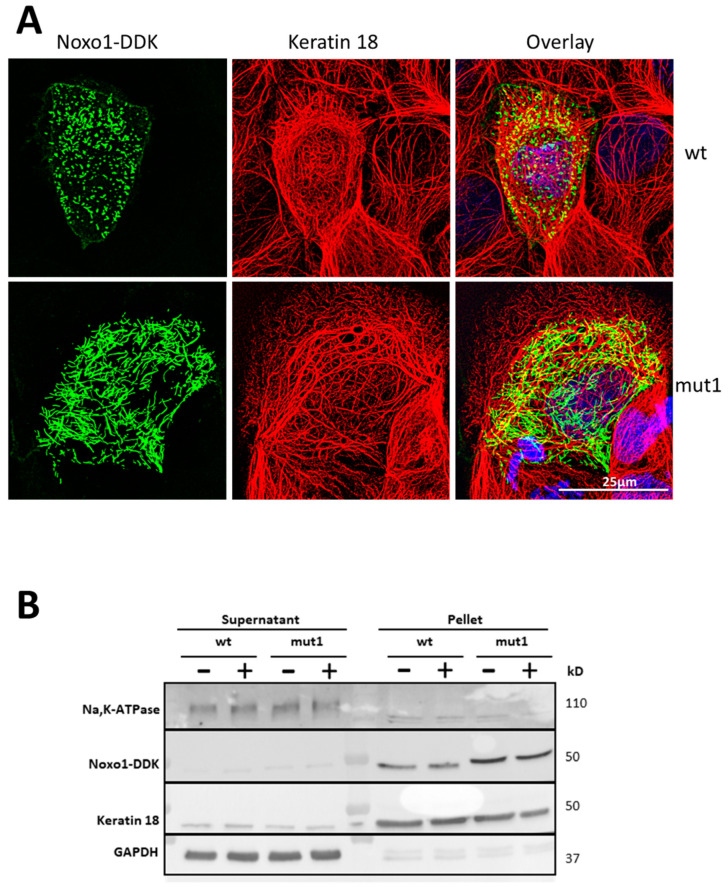
Immunofluorescence Noxo1 with keratin 18 in Caco-2 transfected with wt Noxo1- or mut1 Noxo1-DDK. (**A**) Immunofluorescence wt Noxo1-DDK (left upper panel, green) and keratin 18 (middle upper panel, red). Keratin 18 organization in cells transfected with wt Noxo1-DDK or mut1 Noxo1-DDK assessed with an antibody directed against keratin 18 (red). The overlay upper right panel corresponds to wt Noxo1-DDK/keratin. Immunofluorescence mut1 Noxo1-DDK (left lower panel, green), and keratin 18 (middle lower panel, red). The overlay (right lower panel) corresponds to mut1 Noxo1-DDK/keratin 18. There is a clear colocalization of Noxo1 with keratin 18 at many spots or sites for either the wt or mut1 form of Noxo1 (yellow). Scale bar in all panels: 25 µm. (**B**) Effect of latrunculin B, F-actin depolymerizing agent on the distribution of Na,K-ATPase, Noxo1, keratin 18 and, GAPDH in soluble and insoluble fractions in Caco-2 overexpressing wt or mut1 Noxo1 lysed in RIPA buffer. The result of the expression of these proteins was analyzed by immunoblot using DDK antibody for Noxo1 and specific antibodies against Na,K-ATPase, keratin 18, or GAPDH. Na,K-ATPase and GAPDH are mainly present in the supernatant in all conditions and used as a membrane marker and as loading ctl, respectively. Both wt Noxo1- or mut1 Noxo1-DDK are mainly present in the insoluble fraction (pellet), and to a lesser extent, in the soluble fraction (supernatant). These distributions are independent of whether F-actin is intact or depolymerized in all conditions.

**Table 1 ijms-24-04663-t001:** Antibodies references used during the study.

Antibodies	Species	Reference; Supplier	Dilution for	
			IF	WB
Anti-DDK	Mouse	TA50011-100; Origene Technologies	1/400	1/1000
Anti-DDK	Rabbit	D6w5b mab; Cell signaling	1/400	1/1000
Anti-vinculin	Mouse	V9264; Sigma Aldrich	1/200	1/5000
Anti-Noxa1	Mouse	SC398873; SantaCruz Biotechnologies	1/50	1/500
Anti-Nox1	Goat	Ab121009; Abcam		1/5000
Anti-cyclin d1	Rabbit	2978; Cell signaling		1/500
Phalloidin		P1951; Sigma Aldrich	1/2000	
Anti-calreticuline	Mouse	ADISPA601; Enzo Life Sciences	1/200	
Anti-PDI	Mouse	ADISPA891; Enzo Life Sciences	1/400	
Anti-Vimentin	Rabbit	R28 3932; Cell Signaling	2/200	
Anti Alpha-tubulin	Rabbit	Ab18251; Abcam	1/800	
Anti-Keratin kit	Rabbit and Mouse	9384; Cell signaling		
Anti Citrate synthase	Mouse	SC390693; SantaCruz Biotechnologies	1/400	
Anti-GAD 67	Mouse	MAB5406; Sigma Aldrich	1/500	
Anti-Synaptophysin	Mouse	01011; Synaptic Systems	1/500	
Anti-GFP	Rabbit	SC9996; SantaCruz Biotechnologies		1/500
Anti-GAPDH	Mouse	G8795; Sigma Aldrich		1/5000
Anti-Keratin 18	Mouse	Keratin 18 (DC10), Cell Signaling	1/800	1/2000
Anti-Na,K-ATPase	Mouse	Ab7671; Abcam	1/100	1/2000
Alexa Fluor 488 anti-mouse	Goat	A-11029; Invitrogen	1/800	
Alexa Fluor 546 anti-rabbit	Goat	A-11010; Invitrogen	1/800	
Alexa Fluor 594 anti-rabbit	Goat	Ab15008; Abcam,	1/800	
Alexa Fluor 647 anti-mouse	Goat	Ab150115; Abcam	1/800	
HRP anti-mouse	Horse	7076S; Cell signaling		1/5000
HRP anti-rabbit	Goat	7074S; Cell signaling		1/2500
HRP anti-goat	Donkey	IR 705-035-003; Jackson Immunoresearch		1/2500

## Data Availability

Not applicable.

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
