# Peer review of "Overexpression of a Novel Noxo1 Mutant Increases Ros Production and Noxo1 Relocalisation"

_ijms, 2023, doi:10.3390/ijms24054663_

Round 1

Reviewer 1 Report

This manuscript focuses on Noxo1, a regulatory subunit of the NADPH oxidase Nox1, which is an important source of reactive oxygen species in the colon. The present study shows that a Noxo1 mutation modifies its activity, subcellular distribution and interaction with the cytoskeleton. Evidence that native Noxo1 is not degraded by the proteasome is useful, even if it is not what the authors initially intended to demonstrate. The experimental work seems to have been performed and interpreted correctly. 

Overexpressed Noxo1 does not seem to be degraded by the proteasome in Caco-2 cells. Have you tried to determine if that is also true for endogenous Noxo1?

Could some of the increase in ROS by mut1 be caused by its effect on mitochondria or some other source? Reconstitution of the oxidase in HEK cells shows that mut1 can increase Nox1 activity. However, it does not prove that the increase in ROS observed in Caco-2 cells after mut1 overexpression is due to Nox1. You may want to test Nox1 siRNA or a Nox1 inhibitor in Caco-2 cells to verify that point.

Noxo1 is usually thought to be a cytosolic subunit. Overexpressed Noxo1 does not appear in the cytosolic fraction of Figure 6. Does this rule out the possibility for endogenous Noxo1? This point could be added to the discussion.

It makes sense to mutate D-boxes located outside known functional domains (PX, SH3, PRR) in Noxo1. The text refers to those regions as “unstructured”. However, this term could be understood as “intrinsically unstructured” regions, without alpha helix or beta strand. Is there evidence for the absence of secondary structure around mut1?

Western blots show that mut1 runs at a higher molecular weight than wt. Were the constructs verified by sequencing? Can the replacement of R and L with A explain the shift?

Please add missing band sizes in some Western blots. Also, please add missing numbers of replicates, experiments (n) and statistics in some figure legends.

In Figure 2A was there a band at the size of GFP in the ctl lane? If so, please modify line 107.

Figure 2B, right panel. Please remove the lines connecting the bars and add wt and mut1 labels.

Line 140. Could mut1 be misfolded and not retained in the ER?

Figure 4 H. The expression of cytotoxicity is confusing: 100% toxicity in ctl and higher than 100% in other samples.

Figure 4E and line 168. Apparently mut1 increases all but the smallest size of mitochondrial puncta. The distribution seems to be shifted up, rather than right.

Figure 5B and line 203. The colocalization is hard to see. You could try quantification using software.

Figure 5 legend. Does the word “revelation” really mean “detection”? A clearer alternative could be: “immunoprecipitation with a Noxa1 antibody, followed by immunoblotting with a DDK antibody”.

Line 279. Do these HEK cells have sufficient endogenous expression of p22phox to support overexpressed oxidase subunits?

Lines 287 and 289. Both increases should be expressed the same way: 30% and 110%.

Figure 7B. Is the last ctl group missing? Which ctl group was used for statistics?

Lines 314, 315 and 328. Nox1 and Noxa1 are not endogenous in HEK293 cells.

Figure 8, lower blot, left side. Was Nox1 overexpressed in the left 4 lanes? If not, what are those two bands in the soluble fractions? The Nox1 label is missing on the left side of the blot.

Figure 9 could be removed since it is superseded by Figure 10.

Line 374. Please remove “or Laemmli (C, D) buffers”.

Line 381. Please clarify.

Figure 10A and line 389. The concentrations of MG132 should be the same.

Line 393. (C and D) should read (A and B).

Figure 11 legend. Please shorten to avoid tedious duplications. Line 414 should read “anti-tubulin antibody (red)”. Line 416 please change “F-actin” to “tubulin”.

Line 519. Nocodazole was not mentioned earlier. Please say “not shown”.

Please provide accession numbers of construct inserts to avoid any confusion. 

Author Response

see attached pdf file

Reviewer 2 Report

General:

The authors´ general aim was to investigate whether the mutations in the D-BOX regions affects the stability of NOXO1. However the only experiment to proof this was the treatment with the proteasome inhibitor MG132. To strengthen this evidence, authors should perform additional experiments:

-          Please provide stability assay to test whether the inserted mutations affect the half life time of the NOXO1. One possibility is to add Cx for 5 10, 15, 30, 60, 120min and test the protein expression via western blot and calculate the degradation rate.

-          In addition, please check the ubiquitination of the different NOXO1 variants. This can be performed by adding the proteasome inhibitor MG132 followed by a NOXO1 IP followed WB for ubiquitin

-          If differences were detectable between wt-NOXO1 and mut1/2-NOXO1 it would interestingly to see whether the interacting of NOXO1 and the ubiquitination related proteins is affected. Authors mentioned such candidates themselves in the introduction, such as HACE1, CYLD and the Cbl.

In some Figures, comparison to the empty control vector is missing, Figure 4( localization of citrate synthase in control vector cells), Figure 5 (localization of NOXA1 in control transfected cells), Figure 6 (localization of Na K ATPase in control transfected cells), Figure 10 (expression and regulation of Cycline d1 by MG132), Figure 11(localization of F-actin and Microtubules in control vector cells), Figure 12 (localization of Keratin 18 in control vector cells). These controls rule out that overexpressing of NOXO1 independent of the variant affects the localization of the other proteins.

The differentiation between “soluble compartments” and “insoluble compartments” is an artificial grouping and is not reflecting the physiological situation in the cell but refers to the solubility in specific detergents. Dependent of the detergent used the composition of these “insoluble compartments” changes. Therefore sentences like in line 561-563 “… and the protein can escape this degradation by going into insoluble compartment” should be avoided. Check the manuscript and change these phrases to a more precise formulation.

You performed a co-staining of your NOXO1 mutants with the ER markers Calreticulin and PDI. As a result you wrote in line 140/141 “This shows that the Noxo1 phenotype induced by mut1 is not due to a misfolding of the protein in the ER”. However, NOXO1 is a cytoplasmic subunit of the NOX1 NADPH oxidase complex as neither p47phox no rp67phox and NOXA1 are endoplasmic reticulum localized proteins. Please change or explain this phrase.

The ratio why Cytochalasin B was used is not well explained. Please describe why you use it. Why you think the actin cytoskeleton is involved in the NOXO1 compartmentalization?

Have you tested the interaction between NOX1 and the different NOXO1 variants?

The manuscript is not well structured. The main idea of the mutation in the D-box region is to test whether its stability is affected. Therefore I would recommend putting the section 2.5 after section 2.1

Figure specific comments:

Figure 3: You pointed out in the introduction that NOX1 is not constitutive active. Why you have not tried a stimulus or activator to activate your ROS generation by NOX1? Lucigenin assay is not very reliable sensor for superoxide generation therefore you should use it in combination with a superoxide scavenger such as SOD or PEG-SOD (for intracellular superoxide generation). In addition to proof that you are measuring NOX1 activity, please silence NOX1 with siRNA or a more specific NOX1 inhibitor (e.g. GKT137831).

Figure 5A: Please provide a quantification of the IP shown in 5A as the input shows different concentrations of both NOXA1 as well as the different NOXO1 variants. From the blot itself one could only conclude that the IP follows the expression pattern of the input.

Figure 5B: Did you perform an IF of endogenous NOXA1 protein or die you overexpress it?

Figure 7B: You describe that both wt-NOXO1 and mut1-NOXO1 are able to induce ROS generation in combination of NOXA1 and NOX1 expression, but you do not mention whether mut1-NOXO1 is significantly more potent than wt-NOXO1. But you mentioned in the beginning of the discussion (line 500/501) that overexpressing of mut1-NOXO1, NOXA1 and NOX1 lead to an increased ROS generation compared to wt-NOXO1, NOXA1 and NOX1 overexpressing. Please change the result part accordingly.

Figure 9: The western blot shown for Cyclin D1 is not corresponding to the evaluation shown in Figure 9B. From the blot one could conclude that Cyclin D1 expression is enhanced in mut1-NOXO1 expressing cells compared to wt-NOXO1 expressing cells. Have you quantified this observation? In addition, in mut1-NOXO1 expressing cells there is no difference visible between MG132 treated and not treated cells, from the blot one could even conclude that MG132 does not have an effect in these cells. Please provide a blot better fitting to the quantification and please discuss this observation of mut1-NOXO1 on the expression of Cyclin D1.

Minor:

In line 291/292 you wrote that “this corresponds to the same order of magnitude in terms of increase as those obtained on Caco-2 and HT29-D4 (Figure 3). However in Figure 3 the ROS is much less for the mut1-NOXO1 overexpression and wt-NOXO1 failed complete to increase ROS expression, therefore this statement is misleading, please adapt.

In the discussion, sentence in lines 553/554 a reference is missing.

Author Response

See our point by point answer in the attached file.

Reviewer 3 Report

The work of Fatima-Zahra Benssouina et al. describes the new aspects of Noxo1 protein interactions with cellular components. Authors create reporter genes containing wild type and mutated Noxo1 gene and search the localization and influence of mutation on reactive oxygen species. Mutation was introduced in the sequence box responsible for interaction of Noxo1 with proteasome. Mutation did not significantly influence the degradation of Noxo1 but it influenced the internal cellular skeletal structures, overexpression of mutant Noxo1 changed the arrangement of intermediate filaments in cells. The work is interesting and sheds new light on regulation of redox state and cytoskeletal structures. It is written with good English and is worth of publishing. Some statements in the text and descriptions of Figures would need improvements.

Below detailed remarks  

Line 103

Authors write: “Using an anti-GFP 104 antibody, we showed the expression of the wt Noxo1-GFP (lane 2), mut1 Noxo1-GFP (lane 105 3) or mut2 Noxo1-GFP (lane 4) proteins at the size of ~70 kDa in Caco-2 transfected cells”

Authors write about creation of Noxo1 mutants and in the next sentence about using anti-GFP antibody. It would be better to explain that it was reporter with GFP (Materials and Methods part of the manuscript is at the end and it also does not clearly mention this)

Fig.2B right

It would be good to add description at the upper part of the figure which pattern belongs to wt Noxo1, and mut1 Noxo1

It is also not clear if these forms were counted in single cell or in populations of cells

Line 129

“These four categories are: punctate, short sticks, long sticks and filaments” What was the criterion and how these forms were counted?

The changes in patterns of Noxo structures were shown only to mut1, the influence of mut2 was not shown

Line 133

“mut1 induces a shift from short sticks pattern to a filaments pattern”.

Again mut2 was not studied or not shown

Line 184-

The criteria for puncta classification are not given clearly, how many cells were studied?

Lines 202-204

The statements are not clear

Lines 212-214

The description of Fig.5 could be better, it is not clear what is shown on a, b and c described with small letters. The explanation of immunoblot should be together with description of fragment A, and best if such explanation appeared on the figure itself instead of small letters a,b,c

Line 234

What is “nor mut1”

Line 256

Description of Fig.6A is not precise, this part does not show solubility but it presents that Noxo1 is found in fractions containing membrane enzyme and cytoskeleton

Line 348

The sentence is not understandable

Fig.9

The picture is hard to decipher (marks on the bars are not clear). I would suggest separate charts for wt and mut (can be smaller) than the bars showing influence of proteasome inhibitor could be marked the same way and the idea of everything would become more clear. Additionally in the description A,B,C and D are mentioned while on the figure only A and B are seen. Information about the asterisks is necessary

Fig. 10

This figure is not properly placed in the text (obviously it was previously one figure together with Fig.9). Other remarks  to Fig 10 are the same as to Fig.9.

 Line 414

Probably should be alpha tubulin

Description should also explain the blue color on the photographs

Line 417-418

The sentence “There is no a clear colocalization of Noxo1 with F-417 actin or microtubules except at few spots for either the wt or mut1 form of Noxo1” is unnecessary. It is repeated in the next line of the text.

Line 569

Mistake: may is to

Line 608

The part about cell transfection is badly and carelessly written. The sentence “Then the cells were collected in single cell suspension by treatment with trypsin-EDTA ……” is not understandable, what is single cell suspension? In line 613 there is an unnecessary dot and closing parenthesis that was not open.

Line 624

The source of the Proteosome Extraction Kit is not given, in Line 598 it was different name of the kit, is it the same?

Line 743-754

The first and many sentences further are not understandable and again carelessly written. Did Authors identify pictures visually or by machine learning, and how ?  The whole paragraph on HCS should be rewritten and made more clear.

Author Response

please see the point by point answer provided in attached file.
